# Control of Cytoskeletal Dynamics by β-Arrestin1/Myosin Vb Signaling Regulates Endosomal Sorting and Scavenging Activity of the Atypical Chemokine Receptor ACKR2

**DOI:** 10.3390/vaccines8030542

**Published:** 2020-09-17

**Authors:** Alessandro Vacchini, Cinzia Cancellieri, Samantha Milanesi, Sabrina Badanai, Benedetta Savino, Francesco Bifari, Massimo Locati, Raffaella Bonecchi, Elena Monica Borroni

**Affiliations:** 1Humanitas Clinical and Research Center–IRCCS, 20089 Rozzano (Mi), Italy; alessandro.vacchini@outlook.com (A.V.); Cinzia.Cancellieri@gmail.com (C.C.); samantha.milanesi@humanitasresearch.it (S.M.); sabrina.badanai@gmail.com (S.B.); benedetta.savino@humanitasresearch.it (B.S.); massimo.locati@humanitasresearch.it (M.L.); 2Experimental Immunology, Department of Biomedicine, University of Basel and University Hospital Basel, CH-4031 Basel, Switzerland; 3Cell Culture Facility Unit, IFOM Istituto FIRC di Oncologia Molecolare, 20139 Milan, Italy; 4Department of Medical Biotechnologies and Translational Medicine, University of Milan, 20090 Milan, Italy; francesco.bifari@unimi.it; 5Cell Adhesion Unit, Division of Neuroscience, IRCSS San Raffaele Scientific Institute and San Raffaele University, via Olgettina 58, 20132 Milano, Italy; 6Humanitas University, Department of Biomedical Sciences, 20090 Pieve Emanuele, Italy

**Keywords:** ACKR2, chemokine, β-arrestin1, cytoskeleton, myosin Vb

## Abstract

The atypical chemokine receptor ACKR2, formerly named D6, is a scavenger chemokine receptor with a non-redundant role in the control of inflammation and immunity. The scavenging activity of ACKR2 depends on its trafficking properties, which require actin cytoskeleton rearrangements downstream of a β-arrestin1-Rac1-PAK1-LIMK1-cofilin-dependent signaling pathway. We here demonstrate that in basal conditions, ACKR2 trafficking properties require intact actin and microtubules networks. The dynamic turnover of actin filaments is required to sustain ACKR2 constitutive endocytosis, while both actin and microtubule networks are involved in processes regulating ACKR2 constitutive sorting to rapid, Rab4-dependent and slow, Rab11-dependent recycling pathways, respectively. After chemokine engagement, ACKR2 requires myosin Vb activity to promote its trafficking from Rab11-positive recycling endosomes to the plasma membrane, which sustains its scavenging activity. Other than cofilin phosphorylation, induction of the β-arrestin1-dependent signaling pathway by ACKR2 agonists also leads to the rearrangement of microtubules, which is required to support the myosin Vb-dependent ACKR2 upregulation and its scavenging properties. Disruption of the actin-based cytoskeleton by the apoptosis-inducing agent staurosporine results in impaired ACKR2 internalization and chemokine degradation that is consistent with the emerging scavenging-independent activity of the receptor in apoptotic neutrophils instrumental for promoting efficient efferocytosis during the resolution of inflammation. In conclusion, we provide evidence that ACKR2 activates a β-arrestin1-dependent signaling pathway, triggering both the actin and the microtubule cytoskeletal networks, which control its trafficking and scavenger properties.

## 1. Introduction

Chemokines represent a family of inflammatory mediators with a prominent role in leukocyte migration and angiogenesis, exercised through the activation of dedicated G protein-coupled seven transmembrane domain receptors (7TMRs) [1,2]. Chemokines and their receptors are also involved in cancer biology since, being expressed by both cancer and stromal cells, they dictate the composition of the tumor microenvironment and also directly influence cancer cell proliferation and metastasis [3]. For these biological activities, they represent possible targets for innovative immunotherapies [4]. Chemokines induce cell migration by activating conventional chemokine receptors, which mediate signaling events sustained by G_i_ proteins. A set of atypical chemokine receptors (ACKRs) with no G_i_-mediated signaling activity and devoid of chemotactic activity have been described [5,6]. At present, the ACKR subgroup includes four members: ACKR1 (also known as DARC) [7], ACKR2 (also known as D6) [8], ACKR3 (also known as CXCR7) [9], and ACKR4 (also known as CCX-CKR) [10]. Evidence also suggests that C5L2 [11] and CCRL2 [12] may function similarly to ACKRs. Though being unable to directly support cell migration, ACKRs have non-redundant functions during immune responses, controlling leukocyte mobilization from the bone marrow, their tissue recruitment, and traffic to draining lymph nodes. Evidence indicates that ACKRs fulfil their biological functions by contributing to the generation and maintenance of functional chemokine patterns in tissues by means of different biochemical properties, including the removal, transport, or concentration of their cognate ligands [13].

ACKR2 has served as a model for the definition of ACKRs’ functions in vitro and in vivo [14]. ACKR2 binds with high affinity and promotes the degradation of 13 inflammatory CC chemokines [15]. ACKR2 is expressed at high levels by endothelial cells of lymphatic vessels in several tissues [16], on invading trophoblast and syncytiotrophoblast cells in the placenta [17], and at low levels in some leukocyte subsets [18,19]. In line with its function of a scavenger of inflammatory chemokines, ACKR2 acts as negative regulator of inflammation in tissues where it is expressed [20]. In the context of solid tumors, ACKR2 acts as one mechanism of cancer immunoediting by preventing tumor growth, regulating pro-tumoral leukocyte infiltration. Indeed, inactivation of ACKR2 in gene-targeted animals causes tumor-promoting inflammation in the skin and the gastrointestinal tract [21,22] and a similar observation was made in Kaposi’s sarcoma, where ACKR2 expression is downregulated by the oncogenic KRAS pathway [23]. On the contrary, ACKR2 has a pro-tumorigenic role limiting the recruitment and the activation of protective immune cells in intestinal tumors in Apc^Min/+^ mice [24] and in breast and melanoma metastatic models [25,26]. A scavenging-independent activity has been recently reported in apoptotic neutrophils, where ACKR2 expression results in chemokine presentation rather than internalization and degradation that was instrumental for the resolution of inflammation by promoting efficient efferocytosis and shifting macrophages towards a pro-resolving phenotype [27,28]. In homeostatic conditions, ACKR2 is mostly retained in intracellular compartments associated with both early Rab4/5-positive and recycling Rab11-positive endosomes [29], with a minor pool of receptors undergoing constitutive internalization and recycling [30]. At increasing concentrations of chemokines, ACKR2 scavenging activity increases progressively through an acceleration in the rate of Rab11-dependent recycling of the receptor, resulting in increased availability at the plasma membrane to efficiently guide chemokine to degradative compartments within the cell [31]. ACKR2 trafficking properties are markedly affected by agonistic chemokines, which are bound and degraded with high efficiency, whereas other chemokines bind the receptor with high affinity but are not degraded and do not modify its cellular distribution, acting as neutral ligands [15]. Active but not neutral ligands activate the signaling cascade downstream of ACKR2 that results in cofilin phosphorylation through a G protein-independent β-arrestin1-dependent pathway, relying on Rac1, PAK1, and LIMK1. This signal leads to the actin cytoskeleton rearrangements necessary for increasing receptor cell surface expression and efficiency in the chemokine scavenging function [32].

The cytoskeleton and its regulatory molecules are important in protein endocytosis and trafficking [33] and their importance in 7TMR signal transduction has been addressed [34]. Rapid transport of intracellular vesicles is supported by motor proteins along a cytoskeletal network of microtubules and actin filaments [35]. Several observations suggest that the membrane recycling system is finely regulated at a molecular level by motor proteins such as myosin Vb, a motor protein for actin-based transport of recycling endosomes which associates with multiple endocytic compartments and regulates vesicle exit from perinuclear compartments to the plasma membrane along a microtubule network [36,37,38]. Interestingly, evidence suggests that the components of the signaling cascade activated downstream of ACKR2, such as Rac1 and PAK, are also involved in microtubule organization and dynamics [39,40], and LIMK functions as a signaling node that controls the crosstalk between actin and microtubule networks [41]. To date, evidence indicates that ACKR2 trafficking and scavenging are tightly connected and regulated by cytoskeletal dynamics. Therefore, we set out to investigate in detail the molecular mechanisms required for ACKR2 constitutive and adaptive trafficking properties. Here, we report that cytoskeletal dynamics control ACKR2 constitutive endocytosis and its correct sorting into recycling compartments, while the activity of myosin Vb is required to transport Rab11-positive vesicles from the recycling endosomes to the plasma membrane and sustain receptor upregulation downstream of the β-arrestin1 pathway.

## 2. Materials and Methods

### 2.1. Chemicals and Antibodies

Cytochalasin D (actin polymerization inhibitor), latrunculin A (actin polymerization inhibitor), jasplakinolide (actin polymerization stabilizer), nocodazole (microtubule polymerization inhibitor), paclitaxel (microtubule polymerization stabilizer), Rac1 inhibitor NSC23766, and pertussis toxin (PTX, Gαi inhibitor) were from Calbiochem (Merck4Biosciences, Burlington, MA, USA), and staurosporine was from Merck Millipore (Burlington, MA, USA). Lipofectamine 2000 and anti-human Rab11 antibody were from Invitrogen (Thermo Fisher Scientific, Waltham, MA, USA). ^125^I-CCL2 was from PerkinElmer. Recombinant human chemokines, unconjugated anti-human ACKR2 monoclonal antibody, and rat IgG2a isotype control were from R&D Systems. Antibodies for human myosin Vb and Rab4 were from BD Biosciences. Anti-human β-arrestin1, PAK1, and LIMK1 antibodies were from Cell Signaling Technology. Anti-human α-tubulin and vinculin were from Sigma-Aldrich. AlexaFluor 488-conjugated phalloidin, AlexaFluor-conjugated secondary antibodies, and 4′,6-diamidino-2-phenylindole (DAPI) were from Molecular Probes. Horseradish peroxidase (HRP)-conjugated anti-Ig antibodies were from GE Healthcare.

### 2.2. Plasmids

The Rab11 S25N-pEGFP, the myosin Vb tail fragment pEGFP, and the siRNA LIMK1 plasmids were previously described [42,43,44]. CCR5 was expressed using the pEGFP-N1 vector (Clontech, Mountain View, CA, USA).

### 2.3. Cell Culture and Transfection

Chinese hamster ovary (CHO-K1) cells were grown in DMEM/F12 supplemented with 10% fetal calf serum, 100 U/mL penicillin/streptomycin, 25 mM HEPES, and 650 µg/mL G418. Stable CHO-K1/ACKR2 transfectants were obtained as previously described [45], while CHO-K1/CCR5-pEGFP cells were generated by lipofection, selected with G418, and cloned by limiting dilution. Transient transfectants of CHO-K1/ACKR2 cells were generated by lipofection and were collected 24 h later. For siRNA transfection, CHO-K1/ACKR2 cells were treated for 72 h with 50 nM ON-TARGETplus or control pool (scrambled) siRNA (Dharmacon, Lafayette, CO, USA) and lipofectamine RNAiMax according to the manufacturer’s instructions (Thermo Fisher Scientific). Target protein levels were verified by western blotting.

### 2.4. Chemokine Scavenging and ACKR2 Membrane Expression Analysis

Assays were performed as previously described [31]. Briefly, plasma membrane expression of ACKR2 was evaluated after the CHO-K1/ACKR2 cells were incubated for 1 h at 37 °C with 1 µM cytochalasin D, latrunculin A, jasplakinolide, paclitaxel, and 10 µM nocodazole in the presence or absence of 100 nM CCL3L1, followed by incubation with anti-ACKR2 primary antibody and AlexaFluor 647-conjugated anti-rat antibody labeling (5 μg/mL) for 1 h at 4 °C. When constitutive internalization was evaluated, cells were treated with the indicated inhibitors, incubated with anti-ACKR2 primary antibody at 4 °C, and finally shifted to 37 °C for the indicated time, followed by incubation with AlexaFluor 647-conjugated anti-rat antibody at 4 °C. When CHO-K1/ACKR2 transient transfectants were used, cell surface expression and internalization of ACKR2 was evaluated in two distinct gates referred to as the viable pEGFP^neg^ and pEGFP^pos^ cells, respectively. Events from 3 × 10^5^ events of viable cells were acquired by a BD FACSCanto flow cytometer and analyzed by BD FACSDiva software (BD Biosciences, San Jose, CA, USA). ACKR2-mediated chemokine scavenging was evaluated as previously reported. Briefly, after 6 h of cell incubation with 0.1 nM ^125^I-CCL2 and indicated concentrations of unlabeled CCL2, proteins in the supernatants were then precipitated with 20% tricholoracetic acid (Carlo Erba, Cornaredo MI, Italy) and the radioactivity present in both soluble and insoluble fractions was measured. Degradation rate curves were obtained by data fitting with nonlinear regression and interpolation with the Michaelis–Menten equation (GraphPad Prism, https://www.graphpad.com/scientific-software/prism/). When indicated, cells were pre-treated with 10 μM nocodazole and 1 μM paclitaxel for 1 h, or staurosporine (1 μM) for 6h. When the myosin Vb tail fragment was used, chemokine scavenging was evaluated in sorted pEGFP^neg^ and pEGFP^pos^ CHO-K1/ACKR2 transient transfectants. Eighteen hours after sorting, cells were incubated with 10 ng/mL CCL3L1 and chemokine concentration in the supernatant was measured by ELISA (R&D Systems) at the indicated time points.

### 2.5. Immunofluorescence and Confocal Microscopy Analysis

Immunofluorescence analysis and confocal microscopy analysis were performed as previously described [31]. When indicated, 1 µM cytochalasin D, latrunculin A, jasplakinolide, paclitaxel, 10 µM nocodazole, or 200 µM NSC23766 were added to CHO-K1/ACKR2 cells 1 h before being incubated with chemokine, while 1 μM staurosporine was added 6h before. When PTX was used, cells were treated with 100 ng/mL for 16 h before adding chemokine. To detect microtubules, cells were incubated for 1 h at RT with 1 μg/mL anti-α-tubulin. To visualize myosin Vb and Rab11, cells were incubated for 1 h at RT with 2.5 μg/mL anti-myosin Vb and anti-Rab11, respectively. High-resolution images (1024 × 1024 pixels) were acquired sequentially with a 60x/1.4 numerical aperture Plan-Apochromat oil immersion objective on a FV1000 laser scanning confocal microscope (Olympus, Shinjuku, Tokyo, Japan) and assembled using Photoshop software (Adobe Systems, San Jose, CA, USA). Pearson’s coefficient of correlation (PCC) was measured in a selected region of interest representative of the analyzed cell using Imaris-Coloc software (Olympus, Shinjuku, Tokyo, Japan).

### 2.6. Apoptosis Analysis

Cells (2.5 × 10^5^) were plated in 10% FCS medium in six-well plates and then treated with staurosporine (1 μM) for 6 h. The apoptosis rate was analyzed by flow cytometry using an Annexin V and Porpidium Iodide (PI) staining (BD Bioscence) to detect apoptosis (early apoptosis: annexin V^pos^/PI^neg^; late apoptosis: annexin V^pos^/PI^pos^).

### 2.7. Statistical Analysis

Unless otherwise stated, data were analyzed by one-way ANOVA with a Bonferroni post hoc test (GraphPad Prism).

## 3. Results

### 3.1. ACKR2 Constitutive Recycling Is Regulated by Cytoskeletal Dynamics

The cytoskeleton and its regulatory and motor proteins are involved in the transport of intracellular vesicles along microtubule and actin filament networks, thereby also regulating the trafficking of 7TMRs on the plasma membrane and their signaling activity [34]. As we have previously demonstrated that ACKR2 trafficking is required for its scavenging activity [31], we investigated its association with cytoskeleton element dynamics. In agreement with our previous observations [32], confocal microscopy analysis of CHO-K1/ACKR2 cells showed that in resting conditions, ACKR2 mainly localizes in perinuclear compartments and does not colocalize with actin, but its colocalization with actin is increased by pharmacological modulation of actin dynamics (Figure 1A). Conversely, a strong colocalization of ACKR2 with microtubules in perinuclear compartments was observed, and pharmacological alteration of microtubule dynamics promoted its cytoplasmic rather than perinuclear distribution (Figure 1B). To understand the role of cytoskeleton dynamics in ACKR2 constitutive cycling, its membrane expression level and constitutive internalization rate were quantified by flow cytometry after cell treatment with cytoskeletal stabilizing/destabilizing agents. In an internalization assay, cells were treated with the indicated inhibitors, incubated with anti-ACKR2 primary antibody at 4 °C, and finally shifted to 37 °C for the indicated time, followed by incubation with a secondary antibody at 4 °C. Both actin stabilizing and destabilizing agents induced a significant increase in ACKR2 surface expression levels, with an increased trend also observed for latrunculin A (Figure 1C). While cytochalasin D and jasplakinolide affected ACKR2 constitutive internalization, latrunculin A treatment was ineffective, indicating that this drug might increase ACKR2 surface expression, acting specifically on the receptor recycling pathway (Figure 1D). Inhibition of microtubule dynamics by nocodazole also resulted in increased ACKR2 surface expression (Figure 1E), but no significant effects on ACKR2 constitutive internalization were detected (Figure 1F), suggesting that the microtubule network regulates the recycling properties of ACKR2. Taken together, these results indicate that ACKR2 trafficking properties require intact actin and microtubules networks. The dynamic turnover of actin filaments is required to sustain ACKR2 constitutive endocytosis, while both actin and microtubule networks are involved in processes regulating ACKR2 recycling.

### 3.2. Alteration of Cytoskeletal Dynamics Causes ACKR2 Missorting into Recycling Compartments

ACKR2 is sorted to the plasma membrane through the rapid Rab4-dependent pathway and the slow Rab11-dependent pathway [31]. To better characterize the molecular mechanisms associated with the alteration of ACKR2 cycling properties caused by pharmacological disruption of cytoskeletal networks, pEGFP-tagged Rab4 and Rab11 dominant negative mutants (Rab4-S22N and Rab11-S25N, respectively) were transiently transfected in CHO-K1/ACKR2 cells. Rab4-S22N, but not Rab11-S25N, completely abolished latrunculin A-dependent increases in ACKR2 expression on the cell surface (Figure 2A), indicating that its effect involved rapid recycling pathways and was mediated by Rab4 activity. Conversely, the effect of nocodazole on ACKR2 membrane levels was inhibited by transfection with Rab11-S25N but not Rab4-S22N (Figure 2B), indicating that the nocodazole-induced increase in receptor membrane expression was mediated by Rab11 activity and involved slow recycling pathways. Interestingly, confocal microscopy analysis revealed that latrunculin A treatment resulted in a modest increase in ACKR2 colocalization, with the early endosome marker Rab4 preferentially at the plasma membrane, where F-actin aggregates and actin filaments were lost (Figure 2C). Conversely, in nocodazole-treated cells, the disruption of the microtubule cytoskeleton impaired recycling endosome formation, causing a loss of ACKR2 perinuclear distribution and colocalization with Rab11 (Figure 2D). Taken together, these results demonstrate that dynamic actin turnover and intact microtubules are required for ACKR2 constitutive sorting to rapid Rab4-dependent and slow Rab11-dependent recycling pathways, respectively.

### 3.3. The Scavenger Function of ACKR2 Requires Ligand-Induced Rearrangement of Microtubules

We previously reported that agonist-induced mobilization of ACKR2 from intracellular endosomes is required to improve its chemokine-scavenging properties [31], and demonstrated the need for cofilin-dependent actin rearrangement for its scavenger function [32]. As the trafficking of intracellular vesicles relies on the regulatory and motor proteins of the cytoskeleton [34], we here investigated the role of microtubules in ACKR2 trafficking. Confocal microscopy analysis revealed that ACKR2 activation by its agonist CCL3L1 stimulated microtubule reorganization and increased its colocalization with microtubules, shifting it from the perinuclear compartment to the plasma membrane (Figure 3A). Interestingly, opposite to the atypical chemokine receptor ACKR2, the conventional chemokine receptor CCR5 under basal conditions was preferentially found at the plasma membrane, where it colocalized with microtubules, and its activation by CCL3L1 resulted in receptor internalization and accumulation in the perinuclear compartment, without affecting its colocalization with microtubules (Figure 3B). ACKR2 colocalization with microtubules at the plasma membrane was significantly increased only by ACKR2 agonists (CCL3L1, CCL4, CCL22, CCL2), while neutral ligands (CCL3, its N-term processed CCL3(5-68) variant) and CXCL8, which is not recognized by ACKR2 and was used here as a negative control, were ineffective (Figure 3C,D). Treatment with the microtubule depolymerizing agent nocodazole slightly impaired the agonist-dependent increase in ACKR2 membrane expression (Figure 3E). In line with this, nocodazole also inhibited the ACKR2 chemokine degradation rate (Figure 3F). Opposite to this, microtubule stabilization by paclitaxel had no significant effect on ACKR2 membrane expression (Figure 3D) and increased the chemokine degradation rate (Figure 3F). Taken together, these results demonstrate a role for the microtubule network in ACKR2 trafficking and indicate that not only cofilin-mediated rearrangement of the actin cytoskeleton [32] but also the reorganization of the microtubule network are required to support its scavenger activity.

### 3.4. Myosin Vb Sustains ACKR2 Upregulation and Chemokine Degradation

Membrane recycling events are finely tuned by motor proteins. Among these, myosin Vb is an actin-based transporter which interacts with endocytic compartments and contributes to vesicle trafficking to the plasma membrane along the microtubule network [36,37]. Interestingly, myosin Vb was shown to regulate trafficking and receptor-mediated chemotaxis of the conventional chemokine receptor CXCR2 [43]. Confocal microscopy analysis of CHO-K1/ACKR2 cells stained for endogenous myosin Vb showed that under basal conditions, myosin Vb colocalized with ACKR2 preferentially in perinuclear compartments (Figure 4A), while after stimulation with the ACKR2 agonist CCL3L1, myosin Vb was redistributed and increased its colocalization with ACKR2 (Figure 4A). This effect was most evident at the plasma membrane, though ACKR2 association with myosin Vb was also observed in Rab11-positive recycling vesicles in perinuclear compartments and in Rab11-negative early endosomes diffused within the cytoplasm and at the cell periphery. In contrast, myosin Vb did not colocalize with CCR5, neither in basal conditions nor after CCL3L1 engagement (Figure 4B). To investigate the role of myosin Vb in ACKR2 trafficking, CHO-K1/ACKR2 cells were transiently transfected with a pEGFP-tagged myosin Vb tail, which operates as a myosin Vb dominant negative displacing endogenous myosin Vb and disengaging cargos from F-actin filaments [38]. Confocal microscopy analysis showed a similar intracellular distribution pattern of ACKR2 in cells expressing a myosin Vb tail or not under basal conditions, with the receptor maintaining its colocalization with both microtubules (Figure 4C) and Rab11 (Figure 4D) in perinuclear compartments. However, CCL3L1-dependent reorganization of the microtubule cytoskeleton, with increased ACKR2-microtubule colocalization at the plasma membrane (Figure 4C) and receptor mobilization from Rab11-positive recycling vesicles (Figure 4D), was completely abrogated in cells expressing a myosin Vb tail but not in control untransfected cells. Abrogation of myosin Vb activity did not compromise the constitutive endocytosis of ACKR2 (Figure 4E), but resulted in significantly reduced expression of the receptor at the plasma membrane, inhibition of the chemokine-dependent increase in ACKR2 expression (Figure 4F), and was accompanied by a marked impairment in the ability of ACKR2 to degrade chemokines (Figure 4G). Taken together, these results indicate that after chemokine engagement, ACKR2 requires myosin Vb activity to promote its trafficking from Rab11-positive recycling endosomes to the plasma membrane, which sustains its scavenging activity.

### 3.5. ACKR2 Promotes Microtubule Rearrangement through a G Protein-Independent β-Arrestin1-Dependent Pathway

We previously demonstrated that ACKR2 trafficking and scavenging activities require actin cytoskeleton rearrangement events induced by the activation of a Rac1-PAK1-LIMK1 signaling pathway [32]. As components of this signaling cascade are also involved in microtubule dynamics [39,40], and LIMK functions as a signaling node controlling the crosstalk between actin and microtubule networks [41], we assessed the involvement of this signaling pathway in microtubule rearrangement events associated with ACKR2 activation and its colocalization with the microtubule network. Inhibition of Gα_i_ protein activity by PTX treatment had no effect on CCL3L1-induced microtubule ACKR2 colocalization at the plasma membrane, which was conversely significantly inhibited by β-arrestin1 silencing, consistent with our previous observations that ACKR2 is a β-arrestin1-biased signaling receptor and does not signal via the Gα_i_ pathway (Figure 5A,B). Inhibition of each individual element in the Rac1–PAK1–LIMK1 signaling cascade, either pharmacologically or at a molecular level, did not affect ACKR2 colocalization with microtubules under basal conditions but prevented agonist-dependent ACKR2 redistribution at the plasma membrane and receptor colocalization with microtubules (Figure 5C–E for Rac1, PAK1, and LIMK1, respectively), indicating that agonist-induced activation of the β-arrestin1–PAK1–LIMK1 signaling cascade is required for microtubule rearrangement events sustaining ACKR2 scavenger activity.

### 3.6. Apoptosis-Induced Alteration of Actin Dynamics Impairs ACKR2 Internalization and Scavenging

ACKR2 has been reported on apoptotic neutrophils during the resolution of inflammation, with residual ability to internalize that results in cell surface persistence and increased efficiency in inflammatory chemokine presentation rather than degradation [27,28]. Interestingly, the actin-based cytoskeleton is usually disrupted by actin degradation in apoptotic cells to prevent the release of filamentous actin that serves as a danger signal [46]. To gain insight into a more physiological context that supports the previously observed connection between loss of actin-dependent internalization and persistence of ACKR2 expression at the cell surface, we analyzed receptor trafficking and scavenging properties following treatment with the apoptosis-inducing agent staurosporine. Confocal microscopy analysis of CHO-K1/ACKR2 cells showed that ACKR2 expression on the cell membrane is modified in staurosporine-induced apoptotic cells where the actin cytoskeleton is disrupted (Figure 6A). Flow cytometric quantification of ACKR2 expression showed a decreased receptor level on the cell membrane that inversely correlated with apoptosis rate (Figure 6B). The amount of remaining receptor on the cell membrane displayed a significant impairment in constitutive internalization (Figure 6C) that resulted in a reduced chemokine scavenging activity (Figure 6D). Taken together, these results clearly indicate that apoptosis-induced perturbation of the actin cytoskeleton dramatically impacts ACKR2 trafficking properties, moving the receptor away from its degradative function.

## 4. Discussion

ACKR2, previously referred to as D6, is one of the best characterized member of the “atypical” subfamily of chemokine receptors that, unlike conventional chemokine receptors, do not induce directional cell migration due to their inability to activate the G_i_-mediated signaling pathway [14]. Although their biology is still not completely clarified, evidence from gene-targeted animals clearly indicates that ACKRs overall act as negative regulators of inflammation [20]. Inflammatory chemokines can contribute to shaping the immune component of the tumor microenvironment by recruiting tumor-promoting leukocytes such as myeloid cells, polarized Th2 cells, and Tregs, and by promoting M2-like skewing of tumor-associated macrophages [47]. ACKR2 is also consistently involved in tumor progression in the skin and gastrointestinal tract by regulating pro-tumoral leukocyte infiltration [21,22], and plays a pro-tumorigenic role in Apc^Min/+^ mice bearing intestinal tumors [24] and in breast and melanoma metastatic models [25,26]. Thus, understanding the molecular mechanisms supporting the scavenger activity of ACKR2 may pave the way to innovative therapeutic strategies against inflammation and cancer.

Increasing in vitro and in vivo evidence indicates that ACKR biological functions require the activation of β-arrestin-dependent signaling pathways which, depending on the particular ACKR, support their ability to degrade, transcytose, or present their ligands, finally leading to chemokine gradient shaping in tissues [48]. ACKR2 trafficking and scavenging properties are tightly connected and regulated by the binding of different classes of ligands [31]. Several inflammatory CC chemokines act as ACKR2 agonists, profoundly affecting its cycling properties and finally driving it to degradation, while other chemokines, such as protease-inactivated chemokines, act as neutral ligands binding with high affinity to the receptor without influencing its cellular distribution and are not degraded by ACKR2 [15]. We have previously demonstrated that the scavenger activity of ACKR2 requires the activation of the Rac1–PAK1–LIMK1–cofilin signaling cascade via a G protein-independent β-arrestin1-dependent pathway, leading to actin cytoskeleton reorganization, receptor upregulation, and chemokine degradation [32]. Here we demonstrate that microtubules also have a functional relevance in ACKR2 scavenger activity, as they are required for its correct intracellular sorting and recycling to the cell membrane. Additionally, in this context, results also confirm that ACKR2 is characterized by an unbalanced β-arrestin1-biased signaling behavior and provide further evidence for the biological relevance of ACKR2 signaling activity, as the G protein-independent activation of the Rac1–PAK1–LIMK1 pathway, finely coordinating cytoskeletal network rearrangements, is required to sustain its scavenging activity. Though the molecular mechanism involved in LIMK-dependent control of microtubules remains largely undefined, evidence indicates that LIMK controls microtubule dynamics through phosphorylation of tubulin polymerization-promoting protein 1 (TPPP1/p25) [49], which could therefore represent the missing link between LIMK1 and microtubule rearrangements upon ACKR2 activation.

Myosin Vb is a motor protein which finely regulates membrane protein trafficking after sorting endosomes into the recycling compartment along the microtubule network, interacting with different Rab protein active forms [36,37]. Here, we describe a role of myosin Vb as a motor molecule cooperating with Rab11 to sustain both ACKR2 constitutive and ligand-induced trafficking properties, thus allowing the receptor to develop its scavenger activity. Myosin Vb is also required for the chemotactic activity of conventional chemokine receptors [43], indicating that these two receptor families may use the same molecular networks, though they exert different biological activities and diverge in terms of signaling properties. Myosin Vb activation may be triggered either by a calcium-dependent mechanism or by cargo proteins, which pull the globular domains away from the motor domains, allowing the molecule to extend out into its active state [50]. Conventional chemokine receptors activate calcium transients through the G protein-dependent signaling module [51]. On the other hand, ACKR2 activation does not sustain either the G_i_ pathway nor calcium transients [41], and requires the β-arrestin signaling module to sustain its scavenger activity [32]. We hypothesize that signaling mediators downstream of β-arrestin, or β-arrestin itself, could be responsible for myosin Vb activation through a calcium-independent mechanism, possibly involving its scaffolding function for adaptor proteins.

Constitutive internalization and recycling and ligand-dependent receptor upregulation are unique mechanisms exploited by scavenger receptors, including ACKRs [52], to modulate ligand binding and degradation, coping rapidly with the changes occurring in the tissue. ACKR2 displays a distinguished behavior compared to conventional chemokine receptors and appears to be structurally adapted to actively perform chemokine scavenging, being constitutively internalized, recycled, and not downregulated after chemokine engagement [14]. Although ACKR2 constitutive cycling, which sustains ligand concentration-dependent optimization of its scavenger performance, represents a rapid and unique mechanism to control inflammation, the precise molecular mechanisms involved in ACKR2 sorting to the different recycling pathways are presently unknown [31]. Evidence provided here on the microtubule network, coupled with our previous reports on the direct involvement of actin, indicate that ACKR2 constitutive cycling is finely regulated by cytoskeletal network dynamics. These results also raise the question of whether ACKR2 constitutive cycling requires specific signals to the cytoskeleton from a constitutively active ACKR2. In this respect, we consider it unlikely that ACKR2 constitutive cycling is a direct consequence of its passive transport along pre-existing intracellular trafficking routes, as conventional chemokine receptors do not exert this behavior once expressed in the same cellular context [29]. Interestingly, evidence on other G-protein coupled receptors suggest that their accumulation in the recycling endosome is driven by the phosphorylation status of serine/threonine residues in the cytoplasmic tail of the receptor and its association with β-arrestin [53]. Although the phosphorylation status of the ACKR2 cytoplasmic tail is still debated, it has been demonstrated that ACKR2 expression mediates β-arrestin re-localization to the cell periphery [54], and β-arrestin1 knockdown cells showed a widely diffused pattern of ACKR2 in the cytoplasm and a loss of receptor recycling routes [32], indicating that a crosstalk between ACKR2 and β-arrestins is required to ensure receptor constitutive cycling.

The dynamic reorganization of the cytoskeleton is crucial for regulating intracellular transport processes, migration, and the division of cells. Besides these pivotal roles, several reports indicate that the cytoskeleton is also involved in characteristic morphological changes that occur during apoptosis, but the underlying mechanisms and signaling molecules are not identified in detail [55]. Mounting evidence implicates both actin [56] and microtubule [57] networks as both sensors and mediators of the apoptotic process. In particular, the actin cytoskeleton is usually disrupted by actin degradation in apoptotic cells to prevent the release of filamentous actin that serves as a danger signal, while the microtubule cytoskeleton is reformed during the execution phase of apoptosis, forming an apoptotic microtubule network through a biphasic process that firstly involves a rapid depolymerization and is immediately replaced by extensive bundles of closely packed, newly formed tubulin polymers after actin and intermediate filaments are disassembled. Therefore, our previously reported and here proposed role of the cytoskeleton in the fine-tuning of ACKR2 trafficking and scavenging properties provide evidence of a possible molecular mechanism underlying the scavenging-independent activity that has been recently reported for the receptor when expressed on apoptotic neutrophils, which is instrumental for the resolution of inflammation by promoting efficient efferocytosis and shifting macrophages towards a pro-resolving phenotype [27,28].

## 5. Conclusions

In conclusion, here we provide evidence of G protein-independent β-arrestin-dependent signaling events downstream of ACKR2 controlling the microtubule network and cytoskeletal dynamics. We demonstrate that these events require the β-arrestin1–Rac1–PAK1–LIMK1–myosin Vb signaling pathway and that this pathway is required for ACKR2 trafficking properties and consequent scavenger performance. Because of its increased relevance in inflammation and tumor biology, insights into ACKR2 signaling properties may lead to the identification of new therapeutic approaches acting on the regulation of innate and adaptive immune responses. Finally, a better understanding of ACKR signaling properties may provide new insights into the pharmacology of conventional chemokine receptors. In particular, the intrinsic unbalanced signaling through the β-arrestin module makes ACKR prototypic molecules for the rationale design of allosteric modulators of conventional chemokine receptors, aimed at impairing their G_i_-dependent chemotactic activity, preserving the receptor-mediated β-arrestin-dependent scavenger activity.

## Figures and Tables

**Figure 1 vaccines-08-00542-f001:**
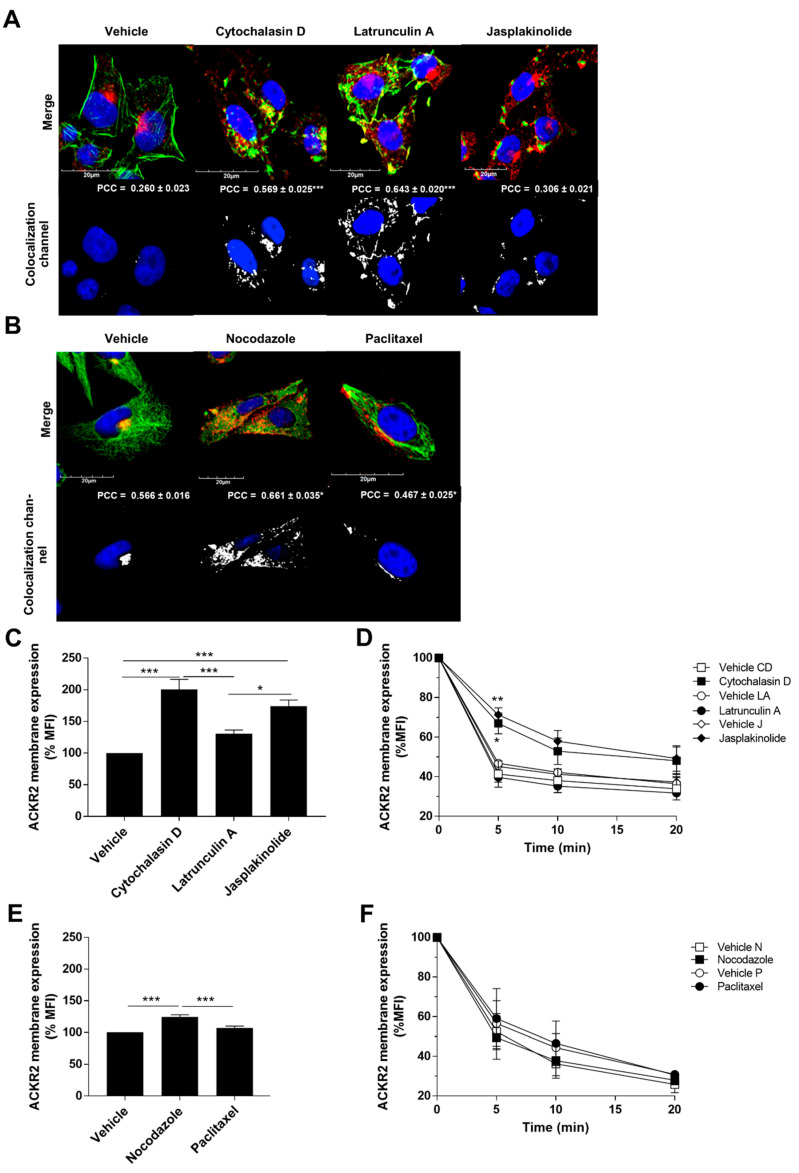
Atypical Chemokine Receptor 2 (ACKR2) constitutive cycling is regulated by cytoskeletal dynamics. (**A**,**B**) Confocal microscopy analysis of Chinese Hamster Ovary (CHO)-K1/ACKR2 cells incubated (30 min) with indicated inhibitors of (**A**) actin and (**B**) microtubule dynamics. Top panels show nuclear staining (DAPI, blue) merged with double staining of ACKR2 (red) and actin (phalloidin staining in panel A; green) or microtubules (α-tubulin staining in panel **B**; green). Bottom panels show colocalization channels of ACKR2 with actin filaments or microtubules, with mean ± SEM of Pearsons’ Correlation Coefficient (PCC) evaluated in at least *n* = 20 cells. Results are normalized over vehicle-treated cells. (**C**–**F**) CHO-K1/ACKR2 cells incubated (30 min) with vehicle (DMSO), 1 µM cytochalasin D (CD), 1 µM latrunculin A (LA), 1 µM jasplakinolide (J), 10 µM nocodazole (N), and 1 µM paclitaxel (P) and analyzed for ACKR2 membrane expression (**A**–**C**,**E**) and ACKR2 constitutive internalization after indicated time points (**D**,**F**). Results are normalized over vehicle-treated cells (panel **C** and **E**) or over 30 min-treated cells with vehicle or inhibitors (panel **D** and **F**), and shown as mean ± SEM of at least *n* = 3 experiments. In panels **D** and **F**, data were analyzed by two-way ANOVA with Tukey’s post hoc test and only statistical analysis of inhibitor versus corresponding vehicle is shown. *: *p* ≤ 0.05, **: *p* ≤ 0.01, ***: *p* ≤ 0.005.

**Figure 2 vaccines-08-00542-f002:**
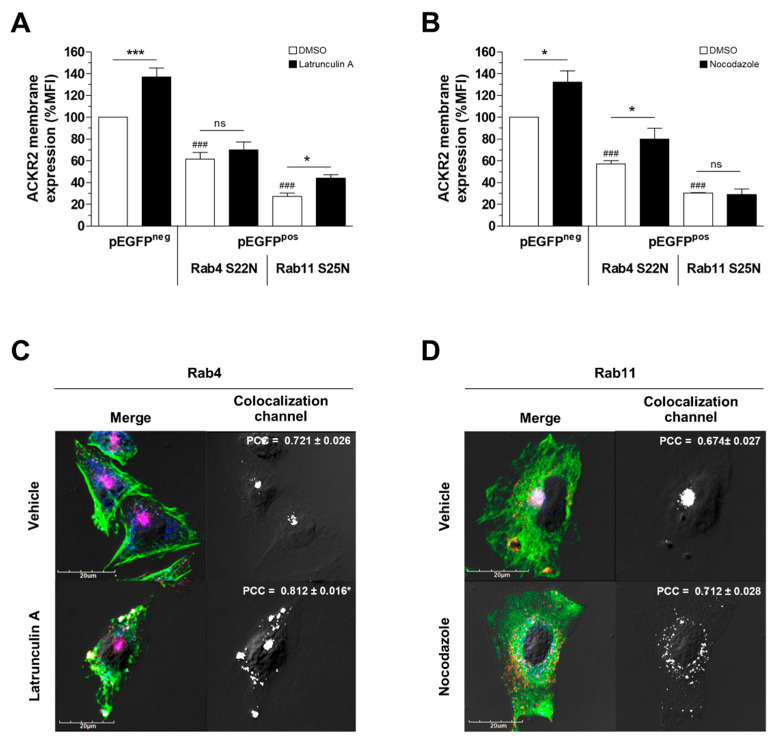
Alteration of cytoskeletal dynamics causes ACKR2 missorting into recycling compartments. (**A**,**B**) Flow cytometry analysis of ACKR2 membrane expression in CHO-K1/ACKR2 cells transiently transfected with the indicated pEGFP-tagged plasmids after incubation (1 h) with vehicle or 1 µM latrunculin A (panel **A**,) and 10 µM nocodazole (panel **B**). Transfected but pEGFP-negative cells (pEGFP^neg^) served as internal control of cells transfected and overexpressing dominant negative Rab4 (Rab4 S22N) and Rab11 (Rab11 S25N) pEGFP plasmids (pEGFP^pos^). Results are representative of mean ± SEM of n = 3 experiments and are shown as percentage of MFI of pEGFP^neg^/pEGFP^pos^ cells. (**C**,**D**) Confocal microscopy analysis of CHO-K1/ACKR2 cells incubated (1 h) with vehicle (DMSO), 1 µM latrunculin A (panel **C**), or 10 µM nocodazole (panel **D**). Left panels show ACKR2 staining (red) merged in panel **C** with Rab4 (blue) and actin (phalloidin staining in green), and in panel **D** with Rab11 (blue) and microtubules (α-tubulin staining in green). Panels on the right show Nomarski interference contrast merged with the colocalization channel. Quantification of ACKR2 colocalization with Rab4 or Rab11 is shown as the mean ± SEM of PCC evaluated in at least *n* = 20 cells. Data were analyzed by two-way ANOVA with Tukey’s post hoc test (panel **A** and **B**), or unpaired Student’s *t*-test (panel **C** and **D**). *: *p* ≤ 0.05, ***: *p* ≤ 0.005 vehicle versus inhibitor-treated cells, ###: *p* ≤ 0.005 pEGFP^pos^ versus pEGFP^neg^ cells.

**Figure 3 vaccines-08-00542-f003:**
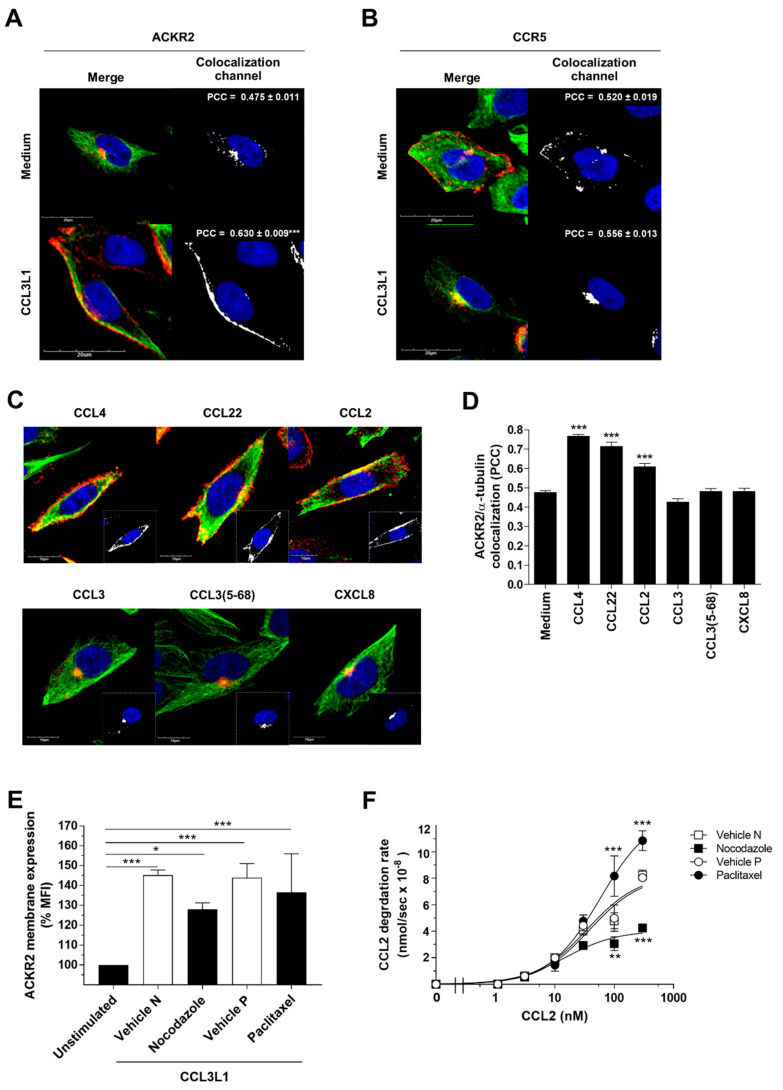
The chemokine scavenger function of ACKR2 requires ligand-induced microtubule rearrangement. (**A**,**B**) Confocal microscopy analysis of CHO-K1/ACKR2 (**A**) and CHO-K1/CCR5 cells (**B**) stimulated with 100 nM CCL3L1 (1 h) and double stained for ACKR2 or CCR5 (red) and microtubule networks (α-tubulin staining; green). Nuclear staining (DAPI) is in blue. Merged images are shown in left panels, right panels show colocalization channels, with quantification of ACKR2 or CCR5 colocalization with microtubules expressed as mean ± SEM of PCC evaluated in at least *n* = 20 cells. (**C**) Confocal microscopy analysis of CHO-K1/ACKR2 upon stimulation with indicated chemokines (100 nM, 1 h). Colors for staining are as in panel A. Colocalization channels are enclosed in the corresponding panels. (**D**) Quantification of ACKR2 colocalization with microtubules in response to treatment of the indicated chemokines. Colocalization results are expressed as mean ± SEM of PCC evaluated in at least *n* = 20 cells. (**E**) Flow cytometry analysis of ACKR2 membrane expression in CHO-K1/ACKR2 cells pre-treated (30 min) with vehicle, 10 µM nocodazole (N), or 1 µM paclitaxel (P) and stimulated with 100 nM CCL3L1 (1 h). (**F**) Effect of microtubule inhibitors (10 µM nocodazole; 1 µM paclitaxel) on ACKR2 scavenging activity in CHO-K1/ACKR2 cells incubated with 0.1 nM 125I-CCL2 and indicated concentrations of CCL2. Results are shown as mean ± SEM of at least *n* = 3 experiments, whereas in panel F of *n* = 2 experiments for paclitaxel treatment. Data were analyzed by unpaired Student’s *t*-test (panel **A** and **B**), or two-way ANOVA with Tukey’s post hoc test (panel **F**). *: *p* ≤ 0.05, **: *p* ≤ 0.01, ***: *p* ≤ 0.005 stimulated versus unstimulated cells (panels **A** to **E**) or vehicle versus inhibitor-treated cells (panel **F**).

**Figure 4 vaccines-08-00542-f004:**
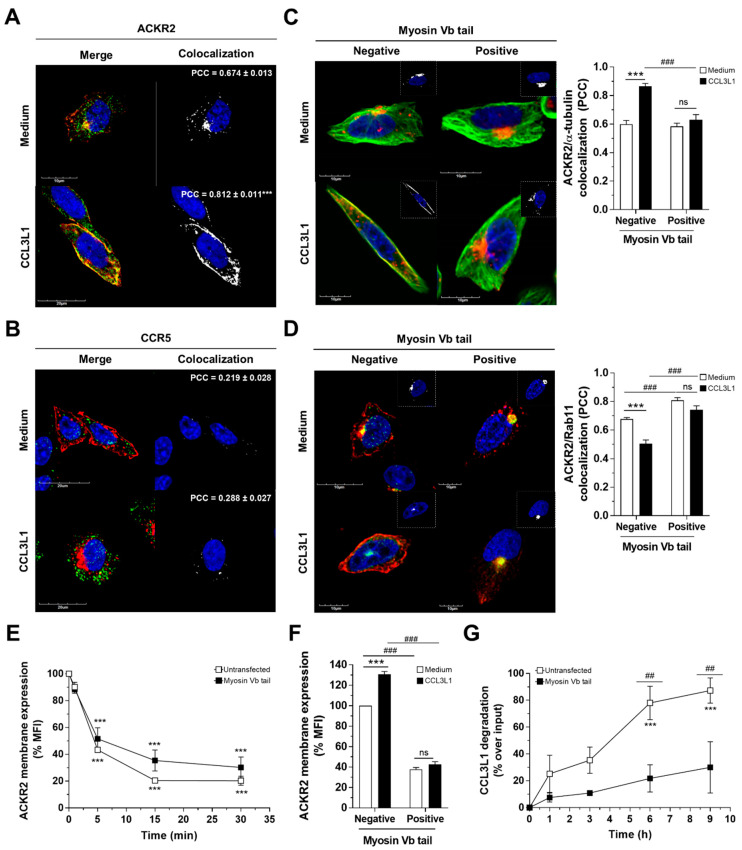
Myosin Vb sustains ACKR2 upregulation and chemokine degradation. (**A**,**B**) Confocal microscopy analysis of CHO-K1/ACKR2 or CHO-K1/CCR5 cells stimulated with 100 nM CCL3L1 (1 h) and double stained for ACKR2 or CCR5 (red; panels **A** and **B**, respectively) and myosin Vb (green). Nuclear staining (DAPI) is in blue. Merged images are shown in left panels, right panels show colocalization channels, with quantification of ACKR2 or CCR5 colocalization with myosin Vb expressed as mean ± SEM of PCC evaluated in at least *n* = 30 cells. (**C**,**D**) Confocal microscopy analysis of CHO-K1/ACKR2 cells transiently transfected with a pEGFP-tagged myosin Vb tail and stimulated with 100 nM CCL3L1 (1 h) and double stained for ACKR2 (red) and microtubules (α-tubulin staining; green; panel **C**) or Rab11 (green; panel **D**). Nuclear staining (DAPI) is in blue. Left and right images refer to cells negative and positive for myosin Vb tail transfection, respectively. Graphs on the right report quantification of ACKR2 colocalization with microtubules (panel **C**) and Rab11 (panel **D**) shown as mean ± SEM of PCC evaluated in at least *n* = 15 cells (open bar: untreated cells, black bar: stimulated cells). (**E**) Flow cytometry analysis of ACKR2 constitutive internalization in transiently transfected pEGFP-negative CHO-K1/ACKR2 cells (open symbols) or expressing a pEGFP-tagged myosin Vb tail (closed symbols), evaluated as membrane expression after incubation with anti-ACKR2 antibody at 4 °C, and finally shifted to 37 °C for the indicated time, followed by incubation with a secondary antibody at 4 °C. (**F**) Flow cytometry analysis of ACKR2 membrane expression in unstimulated (open bars) or stimulated cells for 1 h with 100 nM of CCL3L1. (**G**) ACKR2 scavenging activity evaluated in sorted CHO-K1/ACKR2 cells not expressing (open symbols) or expressing a pEGFP-tagged myosin Vb tail (closed symbols) following incubation with 10 nM CCL3L1 at indicated time points. In panels **E** and **F**, results are shown as mean ± SEM of *n* = 3 experiments, whereas in panel G of *n* = 2 experiments. Data were analyzed by two-way ANOVA with Tukey’s post hoc test (panel **C**–**F**), or unpaired Student’s *t*-test (panel **A** and **B**). ***: *p* ≤ 0.005 stimulated versus untreated cells; ##: *p* ≤ 0.01, ###: *p* ≤ 0.005 negative versus positive, or untransfected versus myosin Vb tail cells.

**Figure 5 vaccines-08-00542-f005:**
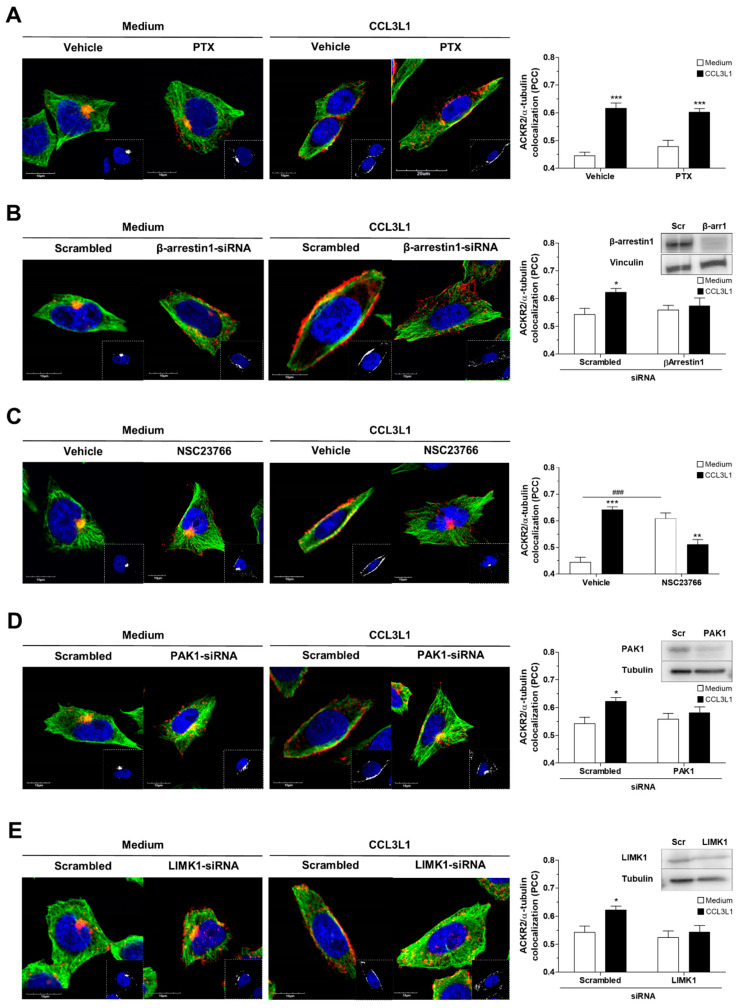
ACKR2 promotes microtubule network rearrangement through a β-arrestin1–Rac1–PAK1–LIMK1 dependent pathway. Confocal microscopy analysis of CHO-K1/ACKR2 cells pretreated with 100 ng/mL PTX (16 h; panel **A**) or 200 µM NSC23766 (1 h; panel **C**), or transfected with scrambled or specific siRNA for β-arrestin1, PAK1, or LIMK1 (50 nM for 72 h in panels **B**, **D**, and **E**, respectively) and stimulated with 100 mM CCL3L1 (1 h). The siRNA-transfected cells were analyzed by western blotting for β-arrestin1, PAK1, and LIMK1 content, as shown in the top right of corresponding panels. Figures show nuclear staining (DAPI) in blue, ACKR2 in red, and microtubules (α-tubulin staining) in green. Quantification of ACKR2 colocalization with microtubules is shown as the mean ± SEM of PCC evaluated in at least *n* = 20 cells. Data were analyzed by two-way ANOVA with Tukey’s post hoc test. *: *p* ≤ 0.05, ***: *p* ≤ 0.01, **: *p* ≤ 0.005 stimulated versus unstimulated cells; ###: *p* ≤ 0.005 vehicle-treated versus NSC23766-treated cells.

**Figure 6 vaccines-08-00542-f006:**
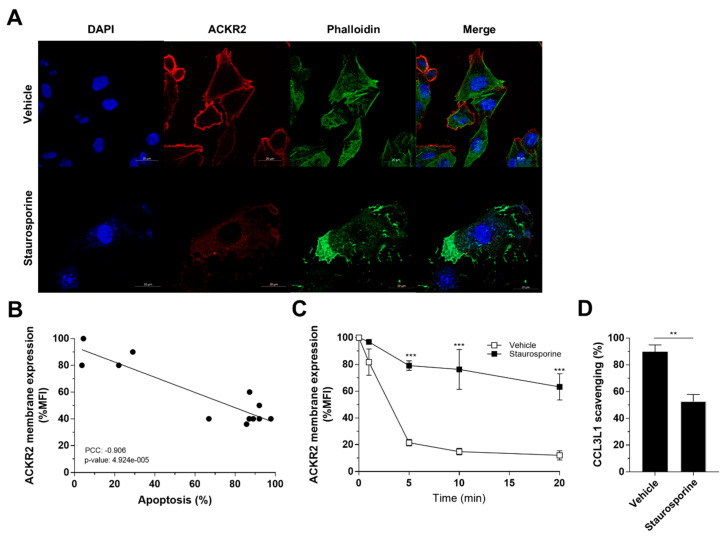
Apoptosis-induced alteration of actin dynamics impairs ACKR2 internalization and scavenging. (**A**) Confocal microscopy analysis of CHO-K1/ACKR2 incubated for 6 h with vehicle (DMSO) or 1 µM staurosporine and double stained for ACKR2 (red) and phallodin (green). Staining was performed in the absence of triton permabilization. Nuclear staining (DAPI) is in blue. CHO-K1/ACKR2 cells were incubated for 6 h with vehicle (DMSO) or 1 µM staurosporine and analyzed for ACKR2 membrane expression (**B**) and ACKR2 constitutive internalization after indicated time points (**C**). Effect of staurosporine (1 µM, 6 h) on ACKR2 scavenging activity in CHO-K1/ACKR2 cells following incubation with 1 nM CCL3L1 for 3 h. Results are shown as mean ± SEM of at least *n* = 3 experiments. Data were analyzed by unpaired Student’s *t*-test (panel **D**) or two-way ANOVA with Tukey’s post hoc test and only statistical analysis of inhibitor versus corresponding vehicle is shown (panel **C**). **: *p* ≤ 0.01, ***: *p* ≤ 0.005.

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
