# Peer review of "Control of Cytoskeletal Dynamics by β-Arrestin1/Myosin Vb Signaling Regulates Endosomal Sorting and Scavenging Activity of the Atypical Chemokine Receptor ACKR2"

_vaccines, 2020, doi:10.3390/vaccines8030542_

Round 1
Reviewer 1 Report
The study by Vacchini et al describes a new mode of regulation for the chemokine scavenging receptor ACKR2. The authors show the constitutive trafficking of ACKR2 and sorting to recycling compartments involves the microtubule network in addition to the actin cytoskeleton. In addition, ACKR2 upregulation and its scavenging properties were shown to be induced by specific agonists in a β-arrestin1-dependent manner that leads to rearrangement of microtubules via myosin Vb activity. Overall, this is a very sound work technically, but the manuscript only elaborates on the mode of activation and redistribution of ACKR2 in vitro. It is currently an incremental work with room for more physiological examination of the proposed regulatory mechanisms that govern ACKR2 function. Such extension of the study will definitely enhance its merits.
Major comments:
- ACKR2 has been shown to be expressed on apoptotic neutrophils during the resolution of inflammation, presumably with minimal internalization and scavenging. However, this expression results in presentation of its ligands and promotion of resolution of inflammation by macrophage reprograming. This information should be added to the manuscript. The actin-based cytoskeleton is usually disrupted by actin degradation in apoptotic cells to prevent the release of filamentous actin that serves as a danger signal. The findings in this manuscript support the connection between loss of actin-based internalization and persistence of ACKR2 expression at the cell surface. To provide a physiological context to their finding and enhance the merit of the manuscript, I strongly advise the authors to examine the regulation of ACKR2 surface expression and scavenging following treatment with apoptosis-inducing agents like FAS ligand or BCL2 antagonist.
- In Fig. 1 panels d and f, the Y axis should be corrected to constitutive internalization. Also, it should be indicated in the legend which time point was used to determine surface expression. In addition, latrunculin is shown to increase surface expression but not reduce internalization. This does not make sense.
Minor comments:
- Line 127, replace and with ,
- The specific target of each inhibitor should be indicated in the methods.
- Please add symbol legend to graphs for clarity.
- The background section in the abstract should be trimmed for more detail on the results.
Author Response
REVIEWER #1
The study by Vacchini et al describes a new mode of regulation for the chemokine scavenging receptor ACKR2. The authors show the constitutive trafficking of ACKR2 and sorting to recycling compartments involves the microtubule network in addition to the actin cytoskeleton. In addition, ACKR2 upregulation and its scavenging properties were shown to be induced by specific agonists in a β-arrestin1-dependent manner that leads to rearrangement of microtubules via myosin Vb activity. Overall, this is a very sound work technically, but the manuscript only elaborates on the mode of activation and redistribution of ACKR2 in vitro. It is currently an incremental work with room for more physiological examination of the proposed regulatory mechanisms that govern ACKR2 function. Such extension of the study will definitly enhance its merits.
Major comments:
1. ACKR2 has been shown to be expressed on apoptotic neutrophils during the resolution of inflammation, presumably with minimal internalization and scavenging. However, this expression results in presentation of its ligands and promotion of resolution of inflammation by macrophage reprograming. This information should be added to the manuscript. The actin-based cytoskeleton is usually disrupted by actin degradation in apoptotic cells to prevent the release of filamentous actin that serves as a danger signal. The findings in this manuscript support the connection between loss of actin-based internalization and persistence of ACKR2 expression at the cell surface. To provide a physiological context to their finding and enhance the merit of the manuscript, I strongly advise the authors to examine the regulation of ACKR2 surface expression and scavenging following treatment with apoptosis-inducing agents like FAS ligand or BCL2 antagonist.
We thank reviewer for his/her comments, and we provide a revised version of the manuscript that includes the reported data on the role of ACKR2 on apoptotic neutrophils and its biological relevance in efferocytosis by promoting resolution of inflammation by macrophage reprograming. During apoptosis, cells undergo characteristic morphological changes in which the cytoskeleton plays an active role, and mounting evidence implicates the both actin [Gourlay CW, Nat Rev Mol Cell Biol. 2005 Jul;6(7):583-9. DOI: 10.1038/nrm1682] and microtubules [Povea-Cabello S, Int J Mol Sci. 2017; Nov 11;18(11):2393. DOI: 10.3390/ijms18112393] networks act as both a sensor and mediator of this process. As suggested by reviewer, to provide a more physiological context to the observations we provided on the cytoskeletal control of ACKR2 activity, we analyzed the effect of apoptosis on receptor trafficking properties and scavenging. Apoptosis can be induced by agents like FAS ligand or BCL2 antagonist, as well as several chemotherapeutic drugs (i.e. cisplatin and staurosporine) have been shown to activate common apoptotic pathways in target cells. Staurosporine operates through both caspase-dependent and caspase-independent pathway depending on the enzymatic equipment of the cell [Belmokhtar CA, Oncogene. 2001 Jun 7;20(26):3354-62. doi: 10.1038/sj.onc.1204436]. Unfortunately, as a consequence of the restrictions due to the Covid-19 emergency, the delivery times of the reagents have been severely slowed down and therefore, in order to ensure the submission of the revised manuscript within the deadline agreed with Editors, we used already available aliquots of staurosporine in our Institute to induce apoptosis that has been checked by Annexin/PI staining on flow cytometry. In this experimental setting, we demonstrated that staurosporine-induced apoptosis modified ACKR2 trafficking properties by reducing receptor expression on cell surface and blocking constitutive internalization, that finally resulted in impaired receptor scavenger function. Since apoptosis blocked ACKR2 constitutive internalization, we would have expected an increased receptor expression on cell surface instead of a decrease. However, the reduced membrane expression of ACKR2 observed could be explained a consequence of apoptosis effects on receptor recycling properties that we have been previously demonstrated been affected by cytoskeletal dynamics (Fig. 2). We provide a revised version of the manuscript that includes a new Figure 6 with the obtained results.
2. In Fig. 1 panels d and f, the Y axis should be corrected to constitutive internalization.
In our experimental setting, constitutive internalization is evaluated by antibody feeding experiment as described in Materials and Methods section (2.4 Chemokine scavenging and ACKR2 membrane expression analysis). A more extensive description of the procedure and data analysis is provided in the following paper “Dissecting Trafficking and Signaling of Atypical Chemokine Receptors” that we published in Methods in Enzymology (Borroni EM, 2013; 521:151-68. DOI: 10.1016/B978-0-12-391862-8.00008-9). Briefly, this technique involves the use of antibodies that recognize receptor-specific extracellular epitopes that can be endogenous signals or engineered extracellular tag on a recombinant receptor. The antibody will label only receptors on the membrane and will follow the receptor until the lysosome if the receptor is targeted to degradation (see Figure 8.1 panel A). The quantification of receptor constitutive internalization is calculated as percent of receptor remaining on the cell membrane (see Figure 8.1 panel B). Thus, in our experimental setting, Y-axis correctly indicates ACKR2 membrane expression.
Also, it should be indicated in the legend which time point was used to determine surface expression.
In panels D and F, surface expression is evaluated at time points indicated in X-axis: 5, 10 and 20 minutes. We add a sentence in the legend as suggested. In panels C and E, surface expression is evaluated immediately at the end of incubation (1 hour) with indicated inhibitors.
In addition, latrunculin is shown to increase surface expression but not reduce internalization. This does not make sense.
According to Reviewer#2 suggestion, we performed a different statistical analysis of our data by using One-way ANOVA with Bonferroni post hoc test instead of unpaired T Student’s test. The new analysis confirmed that cytochalasin D and jasplakinolide induced a significant increase of ACKR2 surface expression levels (% MFI: cytochalasin D = 200.3±16.16; jasplakinolide = 173.9±9.78) but the effect of latrunculin A results no more statistically relevant despite the % MFI is substantially different from vehicle (130.4±6.01). However, the increased trend observed for latrunculin A could be explained as follow. In our previous paper, we demonstrated that ACKR2 undergoes constitutive internalization in clathrin-coated pits vesicles through RAB5-dependent pathway, and constitutive recycling through both rapid, RAB4-dependent, and slow, RAB11-dependent pathways [Bonecchi R and Borroni EM, Blood. 2008]. Latrunculin is an actin-depolymerizing agent that in our Vaccines paper we showed to increase ACKR2 colocalization with actin filament (Fig. 1A), and to regulate the constitutive trafficking of the receptor through shifting towards the RAB4-dependent recycling pathway (Fig. 2A) by increasing ACKR2 colocalization with RAB4-positive vesicles (Fig. 2C). Therefore, although latrunculin does not affect the constitutive internalization of ACKR2 (Fig. 1D), the preferential shifting towards the RAB4-dependent recycling pathway accelerates the mobilization of the receptor from intracellular compartments that could be responsible of the increase of receptor expression on cell membrane (Fig. 1C).
Minor comments:
1.Line 127, replace and with ,
We provide a corrected version of the sentence in the revised manuscript.
2.The specific target of each inhibitor should be indicated in the methods.
We provide a corrected version of paragraph 2.1 Chemicals and antibodies of Materials and methods section with the information about the specific target of each inhibitor.
3.Please add symbol legend to graphs for clarity.
We provide a corrected version of the figures that includes figure legend to graphs.
4.The background section in the abstract should be trimmed for more detail on the results.
We provide a modified version of the background section in the abstract.
Reviewer 2 Report
As background, ACKR2 constitutively recycles through Rab4/Rab11-positive endosomes. This group had previously showed that ACKR2 redistribution requires actin cytoskeleton rearrangements downstream of a b-arrestin1-Rac1-PAK1-LIMK1-cofilin-dependent signaling pathway. Here they demonstrate a role for the microtubule network – more than cofilin phosphorylation. Microtubules required to support myosin Vb-dependent ACKR2 upregulation and scavenging.
The majority of this manuscript relies on the analysis of confocal microscopy of overexpressed receptor and other proteins in CHO-K1 cells, so there are inherent limitations to the approaches used. The images are relatively clear although perinuclear localization can be difficult to differentiate from inclusion bodies or other artifacts. The addition of live cell imaging showing changes in a single cell over time would be more convincing rather than fixed cells before and after treatment with chemokine (understandably this cannot be done when comparing groups treated with different cytoskeletal inhibitors). Orthogonal assays would also make this manuscript stronger as it again relies primarily on analysis of confocal images (although some work with FACS is also performed). Overall the conclusions are supported by the data and do demonstrate roles for different cytoskeletal processes in regulating the recycling of ACKR2
Major Concerns:
- Statistics in the Figure Legends mention that a student’s t-test were used for nearly all comparisons – shouldn’t a one-way ANOVA with Bonferroni correction be used? Notably, in the methods, it is mentioned that two-way ANOVA statistics were used for comparisons.
Minor concerns:
- I think it would be helpful to mention in the main text (and not just the methods) the approach you used to quantify constitutive internalization (labeling at 4 C followed by incubation at 37 C).
- In Fig 1D and F, it would be helpful to have a key next to each graph so the reader does not have to refer to the figure legend to determine which symbol corresponds to each drug. (and this is true for other graphs in the text)
- Would be helpful to mention in the text that Rab4 is associated with rapid recycling and Rab11 with slow recycling pathways to remind readers of the interpretation in Figure 2.
- Figure 4: I worry about the confocal staining in Fig 4D where I do not even observe myosin V staining for CCR5 (4D) with tremendous amounts of staining for ACKR2 (4B).
Author Response
REVIEWER #2
As background, ACKR2 constitutively recycles through Rab4/Rab11-positive endosomes. This group had previously showed that ACKR2 redistribution requires actin cytoskeleton rearrangements downstream of a b-arrestin1-Rac1-PAK1-LIMK1-cofilin-dependent signaling pathway. Here they demonstrate a role for the microtubule network – more than cofilin phosphorylation. Microtubules required to support myosin Vb-dependent ACKR2 upregulation and scavenging.
The majority of this manuscript relies on the analysis of confocal microscopy of overexpressed receptor and other proteins in CHO-K1 cells, so there are inherent limitations to the approaches used. The images are relatively clear although perinuclear localization can be difficult to differentiate from inclusion bodies or other artifacts. The addition of live cell imaging showing changes in a single cell over time would be more convincing rather than fixed cells before and after treatment with chemokine (understandably this cannot be done when comparing groups treated with different cytoskeletal inhibitors). Orthogonal assays would also make this manuscript stronger as it again relies primarily on analysis of confocal images (although some work with FACS is also performed). Overall the conclusions are supported by the data and do demonstrate roles for different cytoskeletal processes in regulating the recycling of ACKR2
We agree with reviewer’s comment about the limitations of the approaches used and the suggestion to shift towards live microscopy in single cells to be more convincing. We have done several efforts also in the past aimed at deciphering ACKR2 trafficking properties using live microscopy in single cells, but we often run into several technical problems mainly due to the stability of the staining of intracellular compartments, that ultimately forced us to work with fixed cells. Differently from fixed conditions in which cells can be easily stained by antibodies, a live setting requires tracers stable over time, resistant to degradation and photobleaching. We generated CHO-K1 cell line stably expressing a pEGFP-tagged version of ACKR2 and we were able to follow and quantify receptor upregulation after chemokine stimulation in live microscopy [Borroni EM, unpublished observation. 2008]. However, when we tried to visualize also intracellular compartment such as Rab4 or Rab11-positive vesicles by generating double transfectants or using fluorescent transferrin as tracer for slow recylcing compartments, and actin filament by adding fluorescent phalloidin, we run into several technical problems, including photobleaching and compensation for fluorochrome overlay. So, we decided to run our experiment in fixed cells and confirm results obtained by other techniques such as cytofluorimetry after treatment with inhibitors (i.e. Rab4/Rab11 dominant negative) [Borroni EM, Methods in Enzymology. 2013]. By using this experimental approach, we were able to decipher and published the trafficking routes of ACKR2 [Bonecchi R and Borroni EM, Blood. 2008], and the involvement of actin [Borroni EM, Sc Signal. 2013] and microtubules [Vaccines paper] cytoskeleton in trafficking properties and scavenging function of the receptor. Therefore, we really appreciate reviewer’s understanding about that, and we thank him/her for stating that our conclusions are still supported by the data provided.
Major Concerns:
Statistics in the Figure Legends mention that a student’s t-test were used for nearly all comparisons – shouldn’t a one-way ANOVA with Bonferroni correction be used? Notably, in the methods, it is mentioned that two-way ANOVA statistics were used for comparisons.
We agree with reviewer concern and according to his/her suggestion, we redid the statistical analysis by replacing unpaired student’s t-test with one-way ANOVA with Bonferroni post hoc test, and we provide detailed description in figure legend of statistical analysis used when one-way ANOVA was not applicable.
Minor concerns:
I think it would be helpful to mention in the main text (and not just the methods) the approach you used to quantify constitutive internalization (labeling at 4 C followed by incubation at 37 C).
We provided a modified version of paragraph “3.1 ACKR2 constitutive recycling is regulated by cytoskeletal dynamics” and the legend of Figure 4, mentioning the approach you used to quantify constitutive internalization
2. In Fig 1D and F, it would be helpful to have a key next to each graph so the reader does not have to refer to the figure legend to determine which symbol corresponds to each drug. (and this is true for other graphs in the text)
We thank the reviewer for his/her comment, and we apologize for the difficulties in figures’ interpretation. We provide a new version of the figures that include legends inside each graph.
3. Would be helpful to mention in the text that Rab4 is associated with rapid recycling and Rab11 with slow recycling pathways to remind readers of the interpretation in Figure 2.
We provided a modified version of paragraph “3.2 Alteration of cytoskeletal dynamics causes ACKR2 missorting to recycling compartments”, mentioning that Rab4 is associated with rapid recycling and Rab11 with slow recycling pathways.
4. Figure 4: I worry about the confocal staining in Fig 4D where I do not even observe myosin V staining for CCR5 (4D) with tremendous amounts of staining for ACKR2 (4B).
Figure 4D shows confocal microscopy analysis of CHO-K1/ACKR2 cells transiently transfected with pEGFP-tagged myosin Vb tail and stimulated with 100 nM CCL3L1 (1 h) and double stained for ACKR2 (red) and Rab11 (green). Nuclear staining (DAPI) is in blue. These colors have been arbitrarily assigned by the operator during image acquisition by confocal microscopy software. Since our experiments were aimed to analyze the colocalization between ACKR2 and tubulin (green in panel 4B) or Rab11, pEGFP staining of myosin Vb tail were only visualized under microscope during acquisition to discriminate between negative and positive cells, thus we did not show pEGFP staining of myosin Vb tail in our representative images. We apologize if the reviewer has been confused by green coloring that is commonly associated to pEGFP.
Round 2
Reviewer 1 Report
One minor correction is needed. In Fig. 1D, Y axis- change D6 to ACKR2
Reviewer 2 Report
Have addressed all my concerns.